# Cases of Maxillofacial Trauma Treated at Hospitals in a Large City in Northeastern Brazil: Cross-Sectional Study

**DOI:** 10.3390/ijerph192416999

**Published:** 2022-12-17

**Authors:** Samuel Benson Lima Barreto, Gustavo Garcia Castro, Ceci Nunes Carvalho, Meire Coelho Ferreira

**Affiliations:** Post-Graduation Program in Dentistry, Universidade CEUMA, São Luís 65075-120, Maranhão, Brazil

**Keywords:** maxillofacial trauma, etiology, treatment

## Abstract

Maxillofacial trauma is associated with facial deformation, loss of function, emotional and social impacts, and high financial costs. This study investigated cases of maxillofacial trauma in a large Brazilian city through a cross-sectional study conducted at two public and two private hospitals. Primary data of 400 patients were collected through a questionnaire, clinical examination, and tomography. Statistical analysis at the 5% significance level was performed. Motorcycle accident was the major cause of trauma (41%); the most frequent trauma and treatment were mandibular fracture (24.3%) and surgery (71%), respectively. The female sex was more affected only regarding domestic accidents (*p* = 0.041) and falls (*p* < 0.001). Motorcycle accidents were more prevalent among 20 to 29 year-olds (*p* < 0.001), followed by physical aggression (*p* < 0.001) and sports accidents (*p* = 0.004). Falls were more frequent among 40 to 59 year-olds (*p* < 0.001). Mandibular fracture affected males and 20 to 29 year-olds more and was mainly associated with motorcycle accidents (48.2%) and physical aggression (22.7%) (*p* = 0.008). Nose fracture was more frequent in falls (29.6%), physical aggression (22.5%), and sports accidents (21.1%) (*p* < 0.001). Compound fracture was associated with motorcycle accidents (84.2%; *p* = 0.028). Maxillofacial trauma (mandibular, nasal, and zygomatic fractures) was associated with motorcycle accidents, physical aggression, and falls. Surgical treatment, hospital care, and public services were the most frequent.

## 1. Introduction

Maxillofacial trauma (MFT) is a worldwide public health problem that can result in facial deformation, loss of chewing function, emotional and social impacts (stress, irritability, low self-esteem, and feelings of social exclusion), and high financial costs [1,2,3,4,5,6]. Considering the growing industrialization, urbanization, and changes in population profiles, MFT has become one of the major medical problems in emergency rooms [7,8,9]. 

MFT ranges from a simple tooth fracture to fractures affecting one or more facial bones and skull injuries [1,9,10,11]. The type of fracture varies depending on the anatomy of the injured site as well as the magnitude and direction of the force of impact [8]. According to the literature, facial fractures most commonly affect the mandible, zygomatic bone, and nose [4,11,12,13,14], which are the most exposed regions of the body [15].

Different factors are involved in the etiology of MFT, such as accidents with motor vehicles and bicycles, physical aggression, the practice of sports, work accidents, and domestic accidents [16,17,18]. Some aspects predispose individuals to a greater incidence of MFT, such as drunkenness or the use of drugs while operating a motor vehicle, and the lack or incorrect use of helmets by motorcyclists and bicyclists. The etiological factors vary depending on the demographic profile of each location, cultural differences, lifestyle differences, and socioeconomic issues [6,10,18,19,20,21].

As Brazil is a large country with distinct regional needs, knowledge regarding social and economic contexts is crucial to understanding the mechanisms involved in the etiology of MFT. Such knowledge would enable the establishment of public prevention strategies as well as estimating and planning therapeutic actions needed in the near future. Moreover, few studies report the initial moment at which injured individuals are admitted to hospital. Such data would provide a higher level of evidence by avoiding the bias associated with secondary data obtained from patient charts. 

Therefore, the aim of the present study was to investigate cases of maxillofacial trauma through a cross-sectional study at public and private hospitals in a large city in northeastern Brazil. 

## 2. Materials and Methods

This study received approval from the Human Research Ethics Committee of Ceuma University (certificate number: 3.306.296) and all participants signed a statement of informed consent. A cross-sectional study was conducted with victims of MFT treated at two public and two private hospitals in the city of São Luís, state of Maranhão, Brazil. This study was conducted between February 2019 and January 2020. To be included, the individuals needed to have been admitted to the Oral and Maxillofacial Surgery and Traumatology Department for the treatment of signs and symptoms of MFT and agree to participate. 

The STROBE guideline was used to ensure proper reporting of the study [22]. The sample size was calculated considering the prevalence of MFT of 35.1% [23], with an estimated error of 4 and a 95% confidence level. The initial sample was determined to be *n* = 546. As the population of injured individuals at the Central Medical Hospital (private), Guarás Hospital (private), Traumatology and Orthopedic Hospital (public), and Socorrão II Hospital (public) in 2018 was 11, 18, 27, and 581 individuals, respectively, an adjustment was made for a finite population, obtaining sample sizes of 10, 17, 27, and 301, totaling 355 individuals. Next, 15% was added to compensate for possible dropouts, leading to a final sample of 408 individuals. 

The participants answered a questionnaire addressing demographic–socioeconomic aspects, previous history of trauma, and the external causes of trauma. The questionnaire was completed by the main researcher and 12 residents of the Oral and Maxillofacial Surgery and Traumatology Department of the hospitals prior to clinical care. If the injured individual was unable to answer the questionnaire, his/her accompanier did so. Questions for which the accompanier was unable to answer were answered by the injured individual at a later time. 

Intraoral and extraoral clinical examinations were performed in the dental clinic under artificial light to investigate the type of facial and oral trauma, followed by computed tomography for the diagnostic confirmation. The following other aspects were recorded on the clinical chart: type of care (outpatient or hospitalized); time elapsed until care; type of treatment (conservative or surgical); length of hospital stay; and outcome (discharge or transference to another hospital). The clinical chart was filled out by an auxiliary health worker. 

The research team had undergone training for filling out the questionnaire and clinical chart. The calibration exercise involved the reading of the instruments, improvement of the phrasing of the questions that raised doubts, clinical training regarding the fractures to be investigated, and training in how to approach the individuals. 

The Statistical Package for the Social Sciences (SPSS, version 21.0 SPSS Inc., Chicago, IL, USA) was used for the data analysis. Descriptive statistics, the chi-square test, Fisher’s exact test, and the likelihood ratio test were employed to investigate associations between the external causes and both sex and age group; the type of MFT and both sex and age group; and the type of MFT and external causes (*p* < 0.05).

## 3. Results

The sample was composed of 400 individuals, resulting in a 98% response rate. The number of participants treated at public hospitals was 91% (364/400) and 62.8% (251/400) were residents of urban areas. The male sex accounted for 76.8% (307/400) of the sample and 20 to 29 year-olds were the largest age group (155/400; 38.8%). The most prevalent self-declared race was brown (274/400; 68.5%). Regarding schooling, 84.8% (339/400) had more than 8 years of studying. A total of 68.5% (274/400) had an income of up to two times the Brazilian monthly minimum wage (equivalent to approximately US$488). 

Motorcycle accidents were the major cause of MFT among the participants (41%) and mandibular fracture was the most prevalent (24.3%). Eight and two individuals were under the influence of alcohol when operating motorcycles and automobiles, respectively. Hospital care (84.3%) and surgical treatment (71%) were the most prevalent. The most frequent hospital stay was four to seven days (42.5%) (Table 1).

Table 2 displays the distribution of the external causes according to sex. Females were more affected only with regard to accidents: domestic (three cases) and falls (thirty-one cases) (*p* = 0.041; *p* < 0.001). 

Table 3 displays the distribution of external causes according to age group. Motorcycle accidents were the more prevalent in the 20-to-29-year-old age group (68 cases), followed by acts of physical aggression (49 cases), and accidents due to the practice of sports (12 cases) (*p* < 0.001; *p* < 0.001; *p* = 0.004). Falls were more frequent in the 40-to-59-year-old age group (22 cases) (*p* < 0.001). 

Table 4 displays the distribution of MFT according to sex, revealing that mandibular fracture was greater among males (91 cases). Table 5 displays the distribution of MFT according to age group, revealing that mandibular fracture was more frequent in the 20-to-29-year-old (43 cases) and 30-to-39-year-old (30 cases) age groups. 

Table 6 displays the distribution of MFT according to external causes. Mandibular fracture was mainly the result of motorcycle accidents (53 cases/48.2%) and acts of physical aggression (25 cases/22.7%) (*p* = 0.008). Nose fracture was mainly the result of falls (21 cases/29.6%), physical aggression (16 cases/22.5%), and accidents during the practice of sports (15 cases/21.1%) (*p* < 0.001). Compound fracture was mainly the result of motorcycle accidents (16 cases/84.2%) (*p* = 0.028).

## 4. Discussion

When comparing the findings of epidemiological studies in different populations, it is necessary to consider aspects such as the geographic region of origin, the demographic profile, and the socioeconomic status of the subjects [6,12,18,21,24]. Considering regional differences, the present study is relevant, as it enables the planning of MFT-prevention strategies for the population of origin of the individuals investigated. This study was conducted in the city of São Luís, which is the capital of the state of Maranhão, with an estimated population of 1,101,884 residents and an average human development index of 0.768 [25].

In the present study, the fact that males and young individuals between 20 and 29 years of age were affected more is in agreement with findings described in studies conducted in different regions [4,6,11,13,14,18,21]. For these demographic aspects, motorcycle accidents were the most prevalent external cause of MFT, which is in agreement with data reported in previous studies [4,12,14,15,26]. 

The larger number of cases of MFT occurring in men is a result of their more aggressive nature and greater exposure to risk factors, such as the uncareful operating of a motor vehicle, the practice of contact sports, and a more active social life, as well as the use of drugs and alcohol [3,4,6,17,18,23]. The greater occurrence of MFT among young adults is due to characteristics inherent to this age, such as greater independence and sociability [26].

In the present study, individuals residing in urban areas were more prone to MFT, which is in agreement with findings described in a study conducted by Batista et al. [12]. Although the regional characteristics of the two studies differ, this finding reveals the weight of urban issues. Population density in urban areas is greater than that found in rural areas, which alone can lead to a greater incidence of MFT. 

Motorcycle accidents constituted the main causal factor of MFT in the present investigation, which is in agreement with data described in previous studies [4,11,20]. The greater prevalence of this external cause is related to individual aspects, such as inexperience, imprudence, and a greater velocity [15], as well as economic aspects, such as the lower cost of acquiring a motorcycle which has led to an increase in sales, the greater ease of movement in heavy traffic, and greater parking ease. Currently, this motor vehicle, together with bicycles, has become an important means of transport in rural areas as well [12]. 

Although the greater ease in acquiring motorcycles is a worldwide phenomenon, it is not accompanied by the use or proper use of a helmet [14,27]. Studies conducted in developing countries report the underuse of this safety item as well as its improper use, which leads to an increased exposure to MFT [20,26].

Acts of physical aggression and accidents suffered during the practice of sports followed motorcycle accidents as the most frequent causes of MFT among men and young adults, who are more likely to become involved in fights and physical activities. Physical aggression was the second most frequent cause of mandibular and nose fractures, whereas the practice of sports was the third most frequent cause of nose fractures. These etiological factors involve the use of the arms and hands to perform ascending movements, whether voluntarily or involuntarily, that reach the target either directly or indirectly. 

Unlike what occurred with motorcycle accidents, physical aggression, and the practice of sports, falls affected women and older age groups more, which is in agreement with data described in previous studies [13,26]. The greater occurrence of falls among women may be due to a variety of situations, such as work accidents, domestic accidents, flight from aggression, or even as a result of one’s own aggression [28]. According to Deslandes [28], falls constitute a “wild card” element that acts in synergy with various forms of violence, such as domestic violence, elder neglect, and working conditions that fail to safeguard one’s physical integrity.

The greater occurrence of falls among older age groups may be related to the neglect of caregivers [28] and issues inherent to one’s own physical condition, such as the neuromuscular and motor limitations inherent to aging and a diminished capacity to avoid hazards in the surrounding environment [18,29,30].

The primary cause of mandibular fracture in the present study was motorcycle accidents, which is in agreement with data described in a study involving patients at a reference hospital in the eastern portion of the state of Minas Gerais (Brazil) [12]. The greater frequency of mandibular, nose, and zygomatic fractures is in agreement with findings reported in previous studies [3,6,13,14,19,21]. These are the most exposed regions of the human body and are therefore more prone to fractures [4,6,15]. 

As MFT generally involves complex treatment, hospital care is required for the majority of cases, as occurred in the present study. The factors that determine immediate hospital care or outpatient care are related to the patient’s health status, the treatment required, hospital flow dynamics, and authorization by health insurance companies. Outpatient care is performed in cases for which conservative treatment is indicated or when the patient is able to await the regression of the swelling until his/her general health status is improved enough to undergo surgical procedures. Hospital care is indicated for cases in which conservative procedures are not possible because the trauma requires more complex procedures in order to minimize permanent sequelae and when the physical condition stemming from the trauma places the patient’s life at risk.

Regarding the response to trauma among younger and older adults, older individuals tend to have an increase in morbidity and mortality rates, as they normally exhibit upper airway impairment, reduced lung function, altered cardiovascular homeostasis, and pre-existing diseases [31]. 

Surgical treatment was performed for the majority of individuals and involved miniplates with 1.5, 2.0, or 2.4 mm titanium screws for rigid internal fixation. The choice of system depends on the type of facial fracture. For mandibular fractures, the 2.0 mm system is used, except in cases of comminuted fractures, for which the 2.4 mm system is used. For fractures of the middle and upper thirds of the face, the 1.5 mm system is used. 

For conservative treatment, clinical and tomographic follow-up were performed. Erich arch bars were used in cases of condyle fracture without the need for open surgery, and for dentoalveolar fractures. 

The most frequent length of hospital stay in the present study was four to seven days, which is in agreement with data described in a study involving patients with facial fractures treated in different hospital emergency sectors in the USA, which reported an average hospital stay of six days [2]. The length of hospital stay is related to awaiting the regression of the swelling, the need for pre-operative exams, awaiting the improvement in the patient’s health status, awaiting authorization from the health insurance company to undergo the surgical procedure, and the availability of a surgery room. 

Considering issues of temporary or permanent functional impairment stemming from the trauma and the financial costs involved, it is of the utmost importance to have knowledge on the most common etiological factors in each geographic region as well as the most frequent types of MFT. Such knowledge enables the planning of prevention actions directed at the causal factors of MFT and the allocation of resources to the sectors involved. 

The negative impact that maxillofacial trauma has on the subjects’ daily lives is a reality, whether through aesthetic impairment or through functions such as mouth opening, chewing, and speech. With the establishment of treatments such as maxillofacial bone coaptation, dental replacement, orthodontic treatment in order to realign the dental arches, restoring the shape of fractured teeth, and installing implants and prostheses, it is necessary for professionals and patients to evaluate the aesthetics and masticatory efficiency [32,33].

Unlike many studies published on MFT which use secondary data, primary data were used in the present investigation. These data were collected by the calibrated researchers at the moment in which the individuals were admitted to hospital. Therefore, the present findings have a higher reliability by avoiding the occurrence of bias related to secondary data collected from patient charts. 

## 5. Conclusions

In conclusion, cases of MFT were mainly the result of motorcycle accidents, acts of physical aggression, and falls. The most common were fractures of the mandible, nose, and zygomatic bone. Surgical treatment, hospitalization, and the use of public hospitals were also more frequent. The present findings underscore the need for educational and preventive measures to make young people aware of the consequences of accidents (especially those involving operating motor vehicles) to the bones of the face and head.

## Figures and Tables

**Table 1 ijerph-19-16999-t001:** Distribution of factors related to maxillofacial trauma. São Luís, Brazil, 2019 (*n* = 400).

Variable	*n* (%)
**Previous history of trauma**	
No	380 (95.0)
Yes	20 (5.0)
**External causes**	
Automobile accident	26 (6.5)
Motorcycle accident	164 (41.0)
Work-related accident	8 (2.0)
Physical aggression	80 (20.0)
Accident with animal	5 (1.3)
Domestic accident	4 (1.0)
Struck by vehicle	13 (3.3)
Practice of sports	28 (7.0)
Assault	15 (3.8)
Fall	57 (14.3)
**Type of maxillofacial trauma**	
Frontal fracture	6 (1.5)
Simple frontal fracture	3 (0.8)
Mandibular fracture	97 (24.3)
Simple mandibular fracture	13 (3.3)
Maxillary fracture	37 (9.3)
Simple maxillary fracture	3 (0.8)
Zygomatic fracture	43 (10.8)
Simple zygomatic fracture	4 (1.0)
Zygomatic arch fracture	9 (2.3)
Zygomatic–orbital fracture	17 (4.3)
Zygomatic–maxillary fracture	12 (3.0)
Orbital fracture	15 (3.8)
Nose fracture	69 (17.3)
Simple nose fracture	2 (0.5)
Nasoethmoid–orbital fracture	12 (3.0)
Compound fracture ^£^	19 (4.8)
Laceration ^§^	35 (8.8)
Abrasion ^§^	1 (0.3)
Contusion ^§^	3 (0.8)
**Dental injury**	
Yes	67 (16.7)
No	333 (83.3)
**Type of dental injury**	
Concussion	2 (0.5)
Subluxation	4 (1.0)
Luxation	19 (4.8)
Fracture	36 (9.0)
Avulsion	6 (1.5)
None	333 (83.3)
**Type of care**	
Outpatient	63 (15.8)
Hospital	337 (84.3)
**Time elapsed until care**	
Immediate care	1 (0.3)
1 to 2 h	199 (49.8)
3 to 8 h	148 (37.0)
9 to 96 h	52 (13.0)
**Type of treatment**	
Conservative	116 (29.0)
Surgical	284 (71.0)
**Hospital stay**	
Discharged on same day as initial care	73 (18.3)
1 to 3 days	123 (30.8)
4 to 7 days	170 (42.5)
8 to 21 days	34 (8.5)
**Outcome**	
Discharge	340 (85.0)
Transference	60 (15.0)

^£^ Compound fracture: when bone is projected outside skin; ^§^ Cases in which no bone fracture occurred.

**Table 2 ijerph-19-16999-t002:** Distribution of external causes according to sex. São Luís, Brazil, 2019 (*n* = 400).

	Sex		
External Causes	Female *n* (%)	Male *n* (%)	*p* *
Automobile accident	3 (11.5)	23 (88.5)	0.144
Motorcycle accident	32 (19.5)	132 (80.5)	0.140
Work-related accident	1 (12.5)	7 (87.5)	0.687 ^§^
Physical aggression	12 (15)	68 (85)	0.051
Accident with animal	1 (20)	4 (80)	1.000
Domestic accident	3 (75)	1 (25)	0.041
Struck by vehicle	5 (38.5)	8 (61.5)	0.191 ^§^
Practice of sports	3 (10.7)	25 (89.3)	0.161 ^§^
Assault	2 (13.3)	13 (86.7)	0.536 ^§^
Fall	31 (54.4)	26 (45.6)	<0.001

* Chi-square test; ^§^ Fisher’s exact test.

**Table 3 ijerph-19-16999-t003:** Distribution of external causes according to age group. São Luís, Brazil, 2019 (*n* = 400).

External Causes	Age Group						
	1–12 Years *n* (%)	13–19 Years *n* (%)	20–29 Years *n* (%)	30–39 Years *n* (%)	40–59 Years *n* (%)	60–90 Years *n* (%)	*p* *
Automobile accident	0 (0.0)	0 (0.0)	9 (34.6)	9 (34.6)	4 (15.4)	4 (15.4)	0.218
Motorcycle accident	0 (0.0)	5 (3.0)	68 (41.5)	59 (36.0)	27 (16.5)	5 (3.0)	<0.001
Work-related accident	0 (0.0)	0 (0.0)	1 (12.5)	6 (75.0)	1 (12.5)	0 (0.0)	0.113
Physical aggression	0 (0.0)	8 (10.0)	49 (61.3)	13 (16.3)	7 (8.8)	3 (3.8)	<0.001
Accident with animal	0 (0.0)	2 (40.0)	0 (0.0)	1 (20.0)	2 (40.0)	0 (0.0)	0.096
Domestic accident	0 (0.0)	0 (0.0)	2 (50.0)	1 (25.0)	0 (0.0)	1 (25.0)	0.646
Struck by vehicle	1 (7.7)	0 (0.0)	4 (30.8)	3 (23.1)	2 (15.4)	3 (23.1)	0.084
Practice of sports	0 (0.0)	7 (25.0)	12 (42.9)	8 (28.6)	1 (3.6)	0 (0.0)	0.004
Assault	0 (0.0)	1 (6.7)	3 (20.0)	4 (26.7)	6 (40.0)	1 (6.7)	0.412
Fall	1 (1.8)	7 (12.3)	7 (12.3)	9 (15.8)	22 (38.6)	11 (19.3)	<0.001

* Likelihood ratio test.

**Table 4 ijerph-19-16999-t004:** Distribution of type of maxillofacial trauma according to sex. São Luís, Brazil, 2019 (*n* = 400).

Type of Maxillofacial Trauma	Sex		
	Female *n* (%)	Male *n* (%)	*p* *
Frontal fracture	6 (66.7)	3 (33.3)	0.006 ^§^
Mandibular fracture ^F^	19 (17.3)	91 (82.7)	0.081
Maxillary fracture ^F^	8 (20.0)	32 (80.0)	0.608
Zygomatic fracture ^G^	12 (21.4)	44 (78.6)	0.728
Zygomatic–orbital fracture	3 (17.6)	14 (82.4)	0.772 ^§^
Zygomatic–maxillary fracture	4 (33.3)	8 (66.7)	0.485 ^§^
Orbital fracture	3 (20.0)	12 (80.0)	1.000 ^§^
Nose fracture ^F^	20 (28.2)	51 (71.8)	0.279
Nasoethmoid–orbital fracture	1 (8.3)	11 (91.7)	0.310 ^§^
Compound fracture	3 (15.8)	16 (84.2)	0.582 ^§^
Soft tissue injury ^y^	14 (35.9)	25 (64.1)	0.049

^F^ Simple fractures were combined with fractures of specific bones; ^G^ simple zygomatic fracture and zygomatic arch fracture; ^y^ cases in which no bone fracture occurred; * chi-square test; ^§^ Fisher’s exact test.

**Table 5 ijerph-19-16999-t005:** Distribution of type of maxillofacial trauma according to age group. São Luís, Brazil, 2019 (*n* = 400).

Type of Maxillofacial Trauma	Age Group						
	1–12 Years *n* (%)	13–19 Years *n* (%)	20–29 Years *n* (%)	30–39 Years *n* (%)	40–59 Years *n* (%)	60–90 Years *n* (%)	*p* *
Frontal fracture	0 (0.0)	0 (0.0)	2 (22.2)	4 (44.4)	3 (33.3)	0 (0.0)	0.329
Mandibular fracture ^F^	1 (0.9)	13 (11.8)	43 (39.1)	30 (27.3)	18 (16.4)	5 (4.5)	0.346
Maxillary fracture ^F^	0 (0.0)	2 (5.0)	15 (37.5)	10 (25.0)	8 (20.0)	5 (12.5)	0.743
Zygomatic fracture ^G^	0 (0.0)	2 (3.6)	18 (32.1)	21 (37.5)	14 (25.0)	1 (1.8)	0.079
Zygomatic–orbital fracture	0 (0.0)	0 (0.0)	7 (41.2)	6 (35.3)	3 (17.6)	1 (5.9)	0.680
Zygomatic–maxillary fracture	0 (0.0)	0 (0.0)	6 (50.0)	2 (16.7)	2 (16.7)	2 (16.7)	0.518
Orbital fracture	0 (0.0)	1 (6.7)	7 (46.7)	4 (26.7)	3 (20.0)	0 (0.0)	0.762
Nose fracture ^F^	0 (0.0)	10 (14.1)	25 (35.2)	15 (21.1)	13 (18.3)	8 (11.3)	0.112
Nasoethmoid–orbital fracture	0 (0.0)	0 (0.0)	5 (41.7)	5 (41.7)	1 (8.3)	1 (8.3)	0.624
Compound fracture	0 (0.0)	0 (0.0)	10 (52.6)	5 (26.3)	4 (21.1)	0 (0.0)	0.231
Soft tissue injury	1 (2.6)	2 (5.1)	17 (43.6)	11 (28.2)	3 (7.7)	5 (12.8)	0.179

^F^ Simple fractures were combined with fractures of specific bones; ^G^ simple zygomatic fracture and zygomatic arch fracture; * likelihood ratio test.

**Table 6 ijerph-19-16999-t006:** Distribution of type of maxillofacial trauma according to external causes. São Luís, Brazil, 2019 (*n* = 400).

Type of Maxillofacial Trauma	External Causes										
	Automobile Accident	Motorcycle Accident	Work-Related Accident	Physical Aggression	Accident with Animal	Domestic Accident	Struck by Vehicle	Practice of Sports	Assault	Fall	*p* *
Frontal fracture	0 (0.0)	1 (11.1)	1 (11.1)	1 (11.1)	0 (0.0)	0 (0.0)	1 (11.1)	1 (11.1)	0 (0.0)	4 (44.4)	0.188
Mandibular Fracture ^F^	7 (6.4)	53 (48.2)	2 (1.8)	25 (22.7)	1 (0.9)	0 (0.0)	1 (0.9)	2 (1.8)	8 (7.3)	11 (10.0)	0.008
Maxillary Fracture ^F^	4 (10.0)	15 (37.5)	1 (2.5)	9 (22.5)	0 (0.0)	1 (2.5)	1 (2.5)	0 (0.0)	4 (10.0)	5 (12.5)	0.199
Zygomatic Fracture ^G^	1 (1.8)	31 (55.4)	0 (0.0)	10 (17.9)	1 (1.8)	0 (0.0)	3 (5.4)	1 (1.8)	2 (3.6)	7 (12.5)	0.113
Zygomatic- orbital fracture	1 (5.9)	9 (52.9)	0 (0.0)	2 (11.8)	1 (5.9)	0 (0.0)	1 (5.9)	1 (5.9)	1 (5.9)	1 (5.9)	0.777
Zygomatic- maxillary fracture	0 (0.0)	9 (75.0)	0 (0.0)	0 (0.0)	0 (0.0)	0 (0.0)	1 (8.3)	2 (16.7)	0 (0.0)	0 (0.0)	0.056
Orbital fracture	2 (13.3)	2 (13.3)	1 (6.7)	4 (26.7)	1 (6.7)	0 (0.0)	0 (0.0)	3 (20.0)	0 (0.0)	2 (13.3)	0.161
Nose fracture ^F^	4 (5.6)	7 (9.9)	3 (4.2)	16 (22.5)	0 (0.0)	1 (1.4)	4 (5.6)	15 (21.1)	0 (0.0)	21 (29.6)	<0.001
Nasoethmoid orbital fracture	0 (0.0)	8 (66.7)	0 (0.0)	3 (25.0)	0 (0.0)	0 (0.0)	0 (0.0)	1 (8.3)	0 (0.0)	0 (0.0)	0.380
Compound fracture	1 (5.3)	16 (84.2)	0 (0.0)	1 (5.3)	0 (0.0)	0 (0.0)	0 (0.0)	0 (0.0)	0 (0.0)	1 (5.3)	0.028
Soft tissue injury	6 (15.4)	13 (33.3)	0 (0.0)	9 (23.1)	1 (2.6)	2 (5.1)	1 (2.6)	2 (5.1)	0 (0.0)	5 (12.8)	0.104

^F^ Simple fractures were combined with fractures of specific bones; ^G^ simple zygomatic fracture and zygomatic arch fracture; * likelihood ratio test.

## Data Availability

The data presented in this study are available on request from the corresponding author. The data are not publicly available due to [restrictions, e.g., privacy or ethical].

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
