# Peer review of "Cases of Maxillofacial Trauma Treated at Hospitals in a Large City in Northeastern Brazil: Cross-Sectional Study"

_ijerph, 2022, doi:10.3390/ijerph192416999_

Round 1
Reviewer 1 Report
The main research findings of this paper will be important for the full understanding of the traumatic injury to the orofacial region by the activity in human life. This article will help design specific safeguards to prevent accidental trauma.
Table 1: It would be easier to see if there was a line of space between the sections (blocks) of the list.
Author Response
Response to Reviewer 1 Comments
Point 1: The main research findings of this paper will be important for the full understanding of the traumatic injury to the orofacial region by the activity in human life. This article will help design specific safeguards to prevent accidental trauma.
Response 1: Thank you for your comment.
Point 2: Table 1: It would be easier to see if there was a line of space between the sections (blocks) of the list.
Response 2: A line of space between the sections (blocks) has been inserted.

Reviewer 2 Report
Dear Authors,
The manuscript is well written and the analysis of the etiological factors and the typology of maxillofacial traumas affecting the population of your city is well conducted. Such work can help improve public prevention strategies to reduce the social and economic costs associated with post-traumatic facial disfigurement.
In accordance with what is highlighted in the Literature, also from your work, it emerges how the failure to comply with the highway code, as well as the abuse of alcohol and drugs, make motorcycle accidents the main cause of maxillofacial trauma among young males.
Helmet use by motorcyclists can prevent major trauma to the brain and upper third of the face and recently it has been shown that the type of helmet used is also a relevant factor to consider.
I ask you to insert this bibliographic reference in this regard, as a supplement to what is correctly underlined in lines 39-41 or in lines 182-184 (Colangeli W, Cordaro R, Boschetti CE, et al. Protective Effects of Helmet Type on Facial Injuries. J Craniofac Surg. 2021;32(4):1591-1595.)
Not only the use of the helmet worn by the motorcyclist is important but also its type. The current regulations on the use of helmets must be implemented with the obligatory wearing of a full-face helmet, precisely to prevent trauma to the jaw, which, as you have also highlighted, represents the most exposed bone segment of the jaw in motorcycle injuries, with serious functional and aesthetic implications.
I would also suggest better specifying in table 4 under the heading "Zygomatic fractures" that the 9 "Zygomatic arch fractures" of Table 1 are also included in the total number.
The article will be accepted for publication in the International Journal of ERPH after minor revisions.
Author Response
Response to Reviewer 2 Comments
Point 1:
The manuscript is well written and the analysis of the etiological factors and the typology of maxillofacial traumas affecting the population of your city is well conducted. Such work can help improve public prevention strategies to reduce the social and economic costs associated with post-traumatic facial disfigurement.
Response 1: Thank you for your comment.
Point 2:
In accordance with what is highlighted in the Literature, also from your work, it emerges how the failure to comply with the highway code, as well as the abuse of alcohol and drugs, make motorcycle accidents the main cause of maxillofacial trauma among young males.
Response 2: It is true. Our findings corroborate with the literature.
Point 3:
Helmet use by motorcyclists can prevent major trauma to the brain and upper third of the face and recently it has been shown that the type of helmet used is also a relevant factor to consider.
I ask you to insert this bibliographic reference in this regard, as a supplement to what is correctly underlined in lines 39-41 or in lines 182-184 (Colangeli W, Cordaro R, Boschetti CE, et al. Protective Effects of Helmet Type on Facial Injuries. J Craniofac Surg. 2021;32(4):1591-1595.)
Response 3: The bibliographic reference (Colangeli et al., 2021) was added in line 183.
Point 4:
Not only the use of the helmet worn by the motorcyclist is important but also its type. The current regulations on the use of helmets must be implemented with the obligatory wearing of a full-face helmet, precisely to prevent trauma to the jaw, which, as you have also highlighted, represents the most exposed bone segment of the jaw in motorcycle injuries, with serious functional and aesthetic implications.
I would also suggest better specifying in table 4 under the heading "Zygomatic fractures" that the 9 "Zygomatic arch fractures" of Table 1 are also included in the total number.
Response 4: For Zygomatic fracture (letter G superscripted) the footnote “GSimple zygomatic fracture and zygomatic arch fracture” was added (Table 4). The same footnote was inserted in the tables 5 and 6.
